Infectious hematopoietic necrosis virus: advances in diagnosis and vaccine development

Yong Chean Yeah 1 2
Ong Hui Kian 3
Tang Hooi Chia 2
http://orcid.org/0000-0002-4736-0674 Yeap Swee Keong 4
Omar Abdul Rahman 2
http://orcid.org/0000-0002-5778-4729 Ho Kok Lian 3
http://orcid.org/0000-0001-9452-198X Tan Wen Siang 1 2 wstan@upm.edu.my
1 Department of Microbiology, Faculty of Biotechnology and Biomolecular Sciences, Universiti Putra Malaysia , Serdang, Selangor , Malaysia
2 Laboratory of Vaccines and Immunotherapeutics, Institute of Bioscience, Universiti Putra Malaysia , Serdang, Selangor , Malaysia
3 Department of Pathology, Faculty of Medicine and Health Sciences, Universiti Putra Malaysia , Serdang, Selangor , Malaysia
4 China ASEAN College of Marine Sciences, Xiamen University Malaysia , Sepang, Selangor , Malaysia
Rahman Mohammad Shamsur
Electronic publication date: 2019 Jul 16
Publication date: 2019
Volume: 7
Electronic Location ID: e7151
Received 2019 Jan 21; Accepted 2019 May 20
Copyright: © 2019 Yong et al.
Copyright year: 2019
Copyright holder: Yong et al.
License: This is an open access article distributed under the terms of the Creative Commons Attribution License, which permits unrestricted use, distribution, reproduction and adaptation in any medium and for any purpose provided that it is properly attributed. For attribution, the original author(s), title, publication source (PeerJ) and either DOI or URL of the article must be cited.
License URL: https://creativecommons.org/licenses/by/4.0/

Keywords: Rhabdovirus, Infectious hematopoietic necrosis virus, DNA vaccine, Diagnosis, Vaccine

Funding: Universiti Putra Malaysia GP-IPS/2016/9511000 This study was supported by a Universiti Putra Malaysia Grant (No: GP-IPS/2016/9511000). The funders had no role in study design, data collection and analysis, decision to publish, or preparation of the manuscript.

==============================
The aquaculture of salmonid fishes is a multi-billion dollar industry with production over 3 million tons annually. However, infectious hematopoietic necrosis virus (IHNV), which infects and kills salmon and trout, significantly reduces the revenue of the salmon farming industry. Currently, there is no effective treatment for IHNV infected fishes; therefore, early detection and depopulation of the infected fishes remain the most common practices to contain the spread of IHNV. Apart from hygiene practices in aquaculture and isolation of infected fishes, loss of fishes due to IHNV infection can also be significantly reduced through vaccination programs. In the current review, some of the diagnostic methods for IHNV, spanning from clinical diagnosis to cell culture, serological and molecular methods are discussed in detail. In addition, some of the most significant candidate vaccines for IHNV are also extensively discussed, particularly the DNA vaccines.

Introduction

Infectious hematopoietic necrosis virus (IHNV) is the causative agent for infectious hematopoietic necrosis in salmonid fishes such as salmon and trout, which represent some of the most important species in aquaculture. Production of worldwide farmed salmon and trout exceeded 3 million tons each year which worth over $17.5 billion (Dixon et al., 2016). Due to the high mortality rate of fishes infected by IHNV particularly in younger fishes (up to 90% or more in fry), the viral outbreaks have resulted in significant economic losses (Ahmadivand et al., 2017; OIE, 2018). The first recorded outbreak occurred in 1950s in blueback salmon brood of 1948 (Rucker, 1953). Highly susceptible fish species which often lead to high mortality include rainbow trout and steelhead trout (Oncorhynchus mykiss), Chinook salmon (O. tshawytscha), coho salmon (O. kisutch), sockeye salmon (O. nerka), chum salmon (O. keta), Biwa trout (O. rhodurus), masu salmon (O. masou), and Atlantic salmon (Salmo salar) (Dixon et al., 2016). IHNV was first isolated from sockeye salmon (Wingfield, Fryer & Pilcher, 1969). It is an enveloped, negative-sense single-stranded RNA virus which belongs to the family of Rhabdoviridae, under the genus Novirhabdovirus with a distinct shape of bullet-like structure. The virion has a size of approximately 150–190 nm in length and 65–75 nm in width when observed under an electron microscope (Fig. 1).

Figure 1 Infectious hematopoietic necrosis virus (IHNV) viewed under a transmission electron microscope.

Any reuse of this figure is only permitted with a full citation of the original source: (Dixon et al., 2016) (Original Publisher: BioMed Central).

Current control methods for IHNV rely on the avoidance of exposure. Therefore, thorough disinfection of fertilized eggs with disinfectants such as iodophor solution, and the use of virus-free water for rearing such as that obtained from undergrounds or treated with UV or ozone are crucial in preventing IHNV, especially in the early phase of farmed salmonid (OIE, 2018). As most of the grow-out phase of the fish occurs in marine environments like net-pens, they could be exposed to viruses shedded from the marine fish reservoirs. Syndromic surveillance is a cost effective approach in minimizing the impact of the virus. If fishes that developed symptoms are separated immediately and culled upon confirmation with a PCR method, farm-wide infections could be prevented (Garver & Wade, 2017). In addition, biosecurity measures such as strict regulations in controlling human and vehicle access to and between farm sites; minimizing contacts between farm fish and wild animals with the use of predators and bird nets; appropriate transport and disposal of fish carcasses, offals and blood water; low stress husbandry practices; applications of fallow periods between cycles; and sterilizing equipment on regular basis with disinfectants such as Virkon® Aquatic are also important in preventing IHNV outbreaks (Wade, 2017).

Infectious hematopoietic necrosis virus RNA genome consists of approximately 11 kilobases, which encodes for six viral proteins: nucleoprotein (N), polymerase-associated phosphoprotein (P), matrix protein (M), glycoprotein (G), non-structural protein (NV), and RNA-dependent RNA polymerase (L) (Kurath et al., 1985). The N protein interacts with the viral RNA genome to form the ribonucleoprotein (RNP) complex, which coils into a bullet-shaped structure. The P and L proteins are associated with the RNP, where they play important roles in the transcription of viral mRNAs and genome replication. The M protein lines the inner surface of the host-derived envelope, which glues the RNP and envelope together, and packs them into a bullet-like shape. In addition, the M protein also inhibits the synthesis of host proteins and induces apoptosis (Chiou et al., 2000).

The NV protein is a non-structural protein, which could only be found in infected cells, but not in the virion (Kurath et al., 1985). The NV protein is essential for the pathogenicity of IHNV (Thoulouze et al., 2004). Most recently, Biacchesi et al. (2017) proposed that the NV protein recruits the PPM1Bb protein phosphatase (Mg2+/Mn2+ dependent, 1Bb) to destabilize the innate immune responses of infected fishes (Biacchesi et al., 2017). The G protein, on the other hand, is a class I viral fusion protein which is present in the outermost layer of IHNV. IHNV G proteins form trimeric peplomers, which are responsible for the viral interaction with its host’s receptor (Coll, 1995). The virus is believed to penetrate the host membrane through receptor-mediated fibronectin endocytosis (Bearzotti et al., 1999; Liu & Collodi, 2002; Nita-Lazar et al., 2016). The G protein alone is capable of inducing protective immunity against IHNV infection (Corbeil et al., 1999). Therefore, the G protein has been studied immensely for vaccine development against IHNV.

Current review focuses on the past and recent advances in the diagnosis and vaccine development against IHNV. To the best of our knowledge, there are only three review articles focusing on IHNV (Alonso & Leong, 2013; Dixon et al., 2016; Nishizawa & Yoshimizu, 2017) for the past 10 years. Dixon et al. (2016) reviewed on the epidemiology. Nishizawa & Yoshimizu (2017) reviewed on the epidemiology and virulence changes, as well as the detection and identification of IHNV. Whereas Alonso & Leong (2013) focused their review on patents on DNA vaccines. Another two reviews (Dalmo, 2018; Hølvold, Myhr & Dalmo, 2014) were on the DNA vaccines for fishes including IHNV. However, none has focused on the diagnosis and vaccine development for IHNV.

Survey methodology

Recently published journal articles (within 10 years) were searched using the keyword “IHNV” in “Scopus” and “Pubmed.” The results were screened and relevant articles used as references for this review. In addition, older information was obtained through “Google” search engine with more specific keywords.

Diagnosis of IHNV

Early detection of IHNV is crucial in controlling and preventing the spread of this infectious disease since there is no effective treatment for the viral infection. Preliminary diagnosis of IHNV is often based on observation of clinical signs and behavioral changes in the fishes. The outward clinical signs and behavioral changes of the IHNV infected fishes can be easily recognized and these diagnoses are able to give a presumptive evidence of IHNV infection. However, serological diagnostic methods such as virus neutralization test (VNT) and enzyme-linked immunosorbent assay (ELISA) are still required to confirm the IHNV infection. Molecular diagnostic methods based on PCR and loop-mediated isothermal amplification (LAMP) technologies are generally considered more advanced due to their higher detection sensitivity as compared to the serological methods. These IHNV diagnostic methods will be discussed in detail in the following sections.

Clinical diagnosis

Typically, fishes infected by IHNV will become lethargic. The infected fishes will also show abnormal swimming patterns such as sporadic whirling, spiral swimming, and flashing. Other symptoms that can be observed through the physical appearance include darkening of the skin color, exophthalmia, pale gills and mucoid, distended abdomens, opaque feces casts, and petechial hemorrhages (Fig. 2) (Woo & Cipriano, 2017).

Figure 2 Clinical signs of IHNV infected fishes.

The infected fishes often show (A) darkening of the skin, (B) exophthalmia, and (C) petechial hemorrhages around the eyes, gills, and fins. Any reuse of this figure is only permitted with a full citation of the original source: (Woo & Cipriano, 2017) (Original Publisher: CABI Publishing).

Several reliable clinical methods for the detection and identification of IHNV are based on the gross and microscopic pathology, chemical pathology, tissue imprints, and electron microscopic analysis. Gross pathological signs of infected fishes include pale internal organs such as the liver, kidney, and spleen; distended abdomen with gelatinous substance; exophthalmia; petechial hemorrhages in the muscles and tissues surrounding the organs of the body cavity; and spinal deformities in surviving fishes. Whereas the microscopic pathological signs include necrosis of eosinophilic granular cells in the intestinal wall, and the degenerative necrosis in hematopoietic tissues, digestive tract, kidney, liver, spleen, and pancreas (Schipp, 2012). As IHNV can cause renal damage to infected fishes, it can lead to significant changes in the cellular and chemical blood constituents. By comparing with uninfected fishes, the ill fishes are anemic and leukopenic, with degenerating thrombocytes and leucocytes. A large amounts of cellular debris (necrobiotic bodies) can therefore be observed in the blood (Woo, Leatherland & Bruno, 2011).

In IHNV infected fishes, splenic and renal hematopoietic tissues are the first and most severely affected areas. Therefore, the cytopathic effect (CPE) of the virus can best be observed using tissue imprints prepared from the kidney and spleen. These imprints often show foamy macrophages and necrobiotic debris, indicating IHNV infection (Kibenge & Godoy, 2016). IHNV infection can also be identified through direct observation of virus particles using an electron microscope. The virions can be detected on the cell surface, within cell vacuoles, as well as in the intracellular spaces of the virus-infected cells (OIE, 2018).

Recently, Burbank, Fehringer & Chiaramonte (2017) reported a non-lethal sampling technique through fin clipping in adult steelhead trout, followed by the detection of IHNV with cell culture techniques. This method has been demonstrated to be more efficient than the standard lethal sampling methods, such as spleen and anterior kidney sampling (Burbank, Fehringer & Chiaramonte, 2017). The confirmation test or “gold standard” for IHNV diagnosis is by detecting the virus in cell cultures, followed by diagnosis using immunological and molecular techniques (Barlič-Maganja et al., 2002; Burbank, Fehringer & Chiaramonte, 2017; Crane & Williams, 2008; Winton, 1991; Woo, Leatherland & Bruno, 2011). The presence of IHNV is routinely assessed by observing the development of viral CPE in cell lines such as epithelioma papulosum cyprinid and fathead minnow under a phase-contrast microscope (Dixon et al., 2016). When virus-like structures are observed in cell cultures with the viral CPE using an electron microscope, a further confirmation test with either the serological method, molecular method, or a combination of both methods is required (OIE, 2018).

Serological diagnosis

Serological diagnosis often requires the use of polyclonal or monoclonal antibodies which bind specifically to the pathogen. The classic VNT is time consuming as it takes 2–8 weeks to complete. Nevertheless, VNT is still being used to detect IHNV infection without sacrificing the fish (Jenčič et al., 2014). More rapid tests based on viral antigen recognition, such as the direct and indirect fluorescent antibody tests (FAT/IFAT) (Arnzen et al., 1991; LaPatra et al., 1989; Woo, Leatherland & Bruno, 2011), ELISA (Adams & Thompson, 2011; Kim et al., 2008), peroxidase immunohistochemical and alkaline phosphatase immunocytochemical (APIC) staining (Drolet, Rohovec & Leong, 1993; Yamamoto et al., 1990), and western blotting (Ristow et al., 1993) have been successfully developed. FAT/IFAT and APIC staining are often used to detect the presence of IHNV in infected fishes through immunostaining of tissue imprints or fixed tissue sections. Drolet, Rohovec & Leong (1993) demonstrated that the APIC assay can detect IHNV in fixed tissue samples over a year old (Drolet, Rohovec & Leong, 1993).

Similarly, ELISA, dot blotting, and western blotting are used to confirm the presence of IHNV by detecting the viral components with antibodies which bind specifically to the viral antigens. To further contribute to the serological detection of IHNV, Xu et al. (2016) performed a high throughput screening method by using the flow cytometry to select recombinant antibodies which could be used as potential universal diagnostic reagents. Another rapid detection method which is known as the staphylococcal coagglutination test can be used to diagnose IHNV within 15 min (Bootland & Leong, 1992; Kim, Winton & Leong, 1994). With the aid of a portable light microscope, this method has the potential to be used as an on-site or point-of-care diagnostic test. However, the staphylococcal coagglutination test is rarely used in the past decade, possibly due to advancements in point-of-care diagnosis using molecular methods. Apart from using specific antibodies, nucleic acid hybridization probes labeled with biotin or alkaline phosphatase can also be employed to detect the presence of IHNV genomic materials (Gonzalez et al., 1997).

Molecular diagnosis

The application of molecular diagnosis in clinical microbiology laboratories accelerates the detection and identification of IHNV. Molecular diagnostic methods are generally better than serological methods in terms of sensitivity, as the presence of IHNV genes can be easily amplified with methods such as PCR and LAMP (OIE, 2018). Since IHNV is an RNA virus, reverse transcription- (RT-) PCR is often used to detect the N and G genes of IHNV (Emmenegger et al., 2000; Knüsel et al., 2007). In addition, real-time RT-PCR (qRT-PCR) is also commonly used to detect IHNV. qRT-PCR generally has a lower risk of contamination, greater sensitivity, and exclusion of post-PCR analysis as compared to RT-PCR (Dixon et al., 2016; Woo, Leatherland & Bruno, 2011). More importantly, qRT-PCR is capable of quantifying the viral genome or transcripts, thereby could be used to determine the health status of an infected fish (Overturf, LaPatra & Powell, 2001).

Purcell et al. (2013) developed a universal qRT-PCR targeting the N gene of IHNV, and reported a sensitivity and specificity of 100%. As quantitation with qRT-PCR requires the establishment of a standard curve, the results generated from different laboratories could be different. Therefore, RT-droplet digital PCR (RT-ddPCR) has been employed for quantitative detection of IHNV as an alternative to qRT-PCR (Jia et al., 2017). In addition, Pinheiro et al. (2016) developed a multiplex RT-PCR (mRT-PCR) for simultaneous detection of major viruses that infect rainbow trout, including IHNV. In the following year, Tong et al. (2017) also developed a liquid chip technique for simultaneous detection of IHNV, spring viremia of carp virus (SVCV), and viral hemorrhagic septicemia virus (VHSV) in salmonids, through the use of fluorescence-coded microspheres for hybridization with the RT-PCR products.

Loop-mediated isothermal amplification or RT-LAMP is a powerful diagnostic tool to detect aquaculture diseases as it is rapid and highly sensitive, in which a few copies of cDNA can be amplified by 109 folds in less than an hour (Biswas & Sakai, 2014; Fu et al., 2011; Suebsing et al., 2011b). Suebsing et al. (2011a) demonstrated that RT-LAMP can detect as little as 0.01 fg of RNA extracted from IHNV-infected cells. In addition, RT-LAMP is suitable to be applied as a point-of-care IHNV detection tool as the amplification of DNA does not require an expensive thermal cycle, which is a must in PCR-based methods. One of the advantages of LAMP in detecting IHNV is that it allows direct and rapid visualization of the amplified products with naked eyes due to the formation of magnesium pyrophosphate (white precipitate byproduct generated from LAMP), which indicates a successful amplification of the target genomic region (Dhama et al., 2014; Gunimaladevi et al., 2005). This feature makes it applicable in laboratory and field conditions.

Ideally, methods established to diagnose IHNV should not be limited to laboratories as they can also be applied in farms which involve a large number of samples. These methods have to be simple, user-friendly, specific, sensitive, rapid, and affordable to fish farmers.

Vaccines against IHNV

IHNV has negatively impacted the wild and hatchery-reared salmonid fishes (Rouxel et al., 2016). For the past 30 years, many researchers have tried to develop effective and safe vaccines to control this disease (LaPatra, Lauda & Jones, 1994; Romero et al., 2011). As early as 1989, Engelking & Leong (1989) purified the G protein from the isolated wild-type IHNV and demonstrated that it provided substantial protection to rainbow trout and Kokanee (O. nerka) against IHNV challenge. Five years later, LaPatra, Lauda & Jones (1994) showed that passive immunity against one strain of IHNV cross protected rainbow trout against all other variants. A subsequent study by Roberti, Rohovec & Winton (1998) demonstrated that two neutralization-resistant attenuated IHNV mutants, namely RB-1 and 193-110-4, conferred significant protection against wild-type IHNV in rainbow trout with a relative percentage of survival (RPS) of 95% and 100%, respectively. Advancements in biotechnological techniques ignited a spark of interest among researchers to produce the recombinant IHNV G protein in bacteria and yeasts as potential vaccine candidates against the disease (Table 1). In addition, bioinformatics analysis of the IHNV nucleotide sequences deposited in the GenBank suggested that the mutation sites of IHNV G protein under positive selections as potential recombinant vaccine candidates (LaPatra, Evilia & Winston, 2008). This idea was adopted by Rouxel et al. (2016) who generated a series of live recombinant IHNV via the reverse genetic approach. This study revealed that the N protein sequence has the most important role to play in the attenuation of IHNV virulence, and modifications of the N and G sequences conferred different degrees of protection and immunity. The details of different types of IHNV vaccines reported in literature are summarized in Table 1.

Table 1 Potential antibodies, subunit, attenuated, and inactivated vaccines produced and tested for IHNV.

Types of vaccines	Agent of inactivation	Tested subject	Vaccination strategy	Outcome	Reference	
Purified glycoprotein	–	Rainbow trout and Kokanee	Immersion immunization, ∼50 µg/mL glycoprotein, 30 days	Protection (RPS: 47–83%) against immersion challenge with IHNV (103–106 TCID50/mL water)	Engelking & Leong (1989)	
IHNV neutralizing antibodies	Antibodies neutralization	Rainbow trout	IP, passive immunization	Neutralizing activity produced against one antigenic variant provided cross protection (RPS: 89–100%) against challenged of IHNV with different antigenic variants (104 PFU/mL)	LaPatra, Lauda & Jones (1994)	
Neutralizing monoclonal antibody-selected attenuated IHNV mutants	Monoclonal antibodies neutralization	Rainbow trout and Kokanee	Immersion immunization, 104–105 TCID50/mL, 24 h	Protection (RPS: 12–65%) against wild type virus (105 TCID50/mL)	Roberti, Rohovec & Winton (1998)	
Escherichia coli expressed nucleoprotein and glycoprotein	–	Rainbow trout	Immersion, bacterial lysate (three mg/mL)	Cross protection (RPS: 38–64%) against three strains of IHNV challenge (103–104 TCID50/mL)	Oberg et al. (1991)	
E. coli expressed glycoprotein	–	Rainbow trout	IP, 50 µg/fish	Induced innate immunity (IFN-1 and IFNγ expression) and protection (RPS: ∼70%) against immersion IHNV challenge (103 TCID50/mL)	Verjan et al. (2008)	
Caulobacter crescentus expressed glycoprotein fused to S-layer protein	–	Rainbow trout	IP, 10 pmol of recombinant protein	Protection (RPS: 26–34%) against IHNV challenge (104–105 PFU/mL)	Simon et al. (2001)	
Sf9 cells expressed glycoprotein	–	Rainbow trout	IP, 1.5 × 105 cells/fish or 50 µL of culture supernatant (Sf9 cells cultured at 20 °C)	Protection (RPS: 56% for cells and 43% for culture supernatant) against IHNV challenge (105 PFU/mL)	Cain et al. (1999a, 1999b)	
Aeromonas salmonicida expressed VHSV and IHNV glycoproteins	–	Rainbow trout	Immersion, live or formalin inactivated bacteria (1/10 dilution)	Protection (RPS: 41% for live and 20% for inactivated bacteria) against IHNV challenge	Noonan, Enzmann & Trust (1995)	
E. coli expressed glycoprotein (IHNV-G-GST)	–	Rainbow trout	IP, 10 µg/fish	Specific antibody against IHNV that can transfer from mother fish to fry and protect (RPS: 50%) against IHNV (106 PFU/mL)	Oshima et al. (1996)	
E. coli and yeast-derived glycoprotein by yeast surface display technology	–	Rainbow trout	Oral, 1.6 × 109 yeast cells	Protection (RPS: 45.8%) against IHNV (102 PFU/mL) via activation of adaptive immunity including upregulation of IgM B cells, helper T cells and cytotoxic T cells; production of specific antibodies; and promotion of antiviral genes expression (IFN-1, Mx-1)	Zhao et al. (2016, 2017)	
Attenuated reverse genetic IHNV	Removal of non-structural (NV) protein or exchange to viral hemorrhagic septicemia virus glycoprotein	Rainbow trout	IP, 106 PFU/mL	Protection (RPS: 100%) against IHNV challenge (2 × 106 PFU/mL); without specific antibody production nor promotion of antiviral IFN/IFN related genes	Romero et al. (2008)	
			NV protein promotes nitric oxide and reactive oxygen species production by macrophages which help to protect against the infection	Romero et al. (2011)	
	Modified nucleoprotein (N) and glycoprotein (G) sequence	Rainbow trout	Immersion, 5 × 104 PFU/mL	Different modifications of N and G gene sequences resulted in different protection efficacy against IHNV infection. N2G3 strains provided the best protection (RPS: 86%) against IHNV (5 × 104 PFU/mL) through activation of specific antibody and antiviral system (IFN-1)	Rouxel et al. (2016)	
Synthetic peptides P76, P226, P268	–	Rainbow trout	IP, one mg/fish	Specific antibody against IHNV but inconsistent, no challenge trial	Emmenegger et al. (1997)	
Infectious pancreatic necrosis virus	–	Rainbow trout	IP, 106.3 TCID50	Protection (RPS: 68.8%) against IHNV (105 TCID50)	Kim et al. (2009)	
Polyinosinic polycytidylic acid (poly(I:C))	–	Rainbow trout	IP, 50 µg/fish	Protection (RPS: 95.2%) against IHNV (105 TCID50)	Kim et al. (2009)	
Inactivated vaccines	Binary ethylenimine (BEI), β-propiolactone (BPL), formaldehyde	Rainbow trout	IP, 8 × 105.82 TCID50	Induced specific IgM in serum and mucus (skin surface and gills); BPL inactivated vaccine > PEI inactivated vaccine > formaldehyde inactivated vaccine	Tang et al. (2016)	
Protection against IP IHNV challenge (10 × 103.36 LD50); BPL inactivated vaccine (RPS: 91.67%) > PEI inactivated vaccine (RPS: 83.33%) > formaldehyde inactivated vaccine (RPS: 79.17%)	
BEI, BPL, formaldehyde, heat	Rainbow trout	IP and IM, 107.5 TCID50	BPL inactivated vaccine induced consistent protection against IP IHNV challenge (105 PFU/mL)	Anderson et al. (2008)	
Attenuated IHNV	Tissue culture passage 100×	Rainbow trout	IP, 105 (day 0); 107 (2 months); 2 × 107 (4 months)	Production of specific antibody	Ristow et al. (2000)	
Rainbow trout	IN and IM, 106 PFU/mL	IN provided comparable protection against IM vaccination; live IHNV challenged (5–10,000 PFU/mL) through activation of nasopharynx-associated lymphoid tissue IgT+ B cells without causing damage to central nervous system	LaPatra, Kao & Erhardt (2015), Larragoite et al. (2016), Salinas, LaPatra & Erhardt (2015), and Tacchi et al. (2014)	
Note:

BEI, binary ethylenimine; BPL, β-propiolactone; NV, non-structural protein; N, nucleoprotein; G, glycoprotein; RPS, relative percentage of survival; IN, intranasal delivery; IP, intraperitoneal delivery; IM, intramuscular delivery; TCID50, 50% tissue culture infectious dose; LD50, Lethal dose that kills 50% of subjects; PFU, plaque-forming unit; IFN, interferon; IHNV, infectious hematopoietic necrosis virus.

Despite an intensive development of recombinant vaccines against IHNV, biotechnology-based vaccines such as attenuated vaccines, recombinant subunit vaccines, live recombinant vaccines, and even reverse genetic vaccines are not commercially available, where their developments are encumbered by safety concerns toward consumers and environment (Romero et al., 2011). Thus, more efforts are needed in performing major field trials and commercialization of the potential IHNV vaccines listed in Table 1.

Development of DNA vaccine

DNA vaccine is a type of genetic vaccine which involves the introduction of recombinant plasmid encoding an immunogenic antigen into host cells, whereby the antigen could be translated and primes the immune system (Van Drunen Littel-Van Den Hurk, Loehr & Babiuk, 2001). Along with the advancement in genetic engineering, numerous DNA vaccines have been invented in the past three decades and many have entered clinical trials (Ferraro et al., 2011). Although the developed DNA vaccines are more focused on targeting human diseases, DNA vaccines against IHNV were also frequently reported (Alonso et al., 2003; Ferraro et al., 2011; Tonheim, Bogwald & Dalmo, 2008). An obvious advantage of DNA vaccines over protein-based vaccines is the scalability and lower cost of production with reduced complexities. DNA vaccines could circumvent most of the problematic issues associated with protein-based vaccines including challenges in protein purification, low protein expression, low protein solubility, and protein misfolding (Leitner, Ying & Restifo, 1999). Importantly, most DNA vaccines were also demonstrated to be capable of inducing both the cellular and humoral immune responses similar to the live attenuated vaccines (Wang et al., 1998). Moreover, DNA vaccines have a better safety profile in contrary to live attenuated vaccines comprising attenuated pathogens, which may pose a risk of regaining virulence in the host (Pliaka, Kyriakopoulou & Markoulatos, 2012). In addition, plasmid DNA containing immunostimulatory sequence (CpG motifs) also increases the immunogenicity of the vaccine and reduces the reliance on toxic adjuvants which often result in adverse inflammation (Coombes & Mahony, 2001). Plasmid DNA could also be engineered to encode multiple viral antigens to generate multivalent DNA vaccines (Tonheim, Bogwald & Dalmo, 2008).

Majority of the IHNV DNA vaccines were developed based on the G protein of the IHNV M and U genotypes, which were found to induce strong humoral immune responses in immunized fishes (Nichol, Rowe & Winton, 1995; Peñaranda, LaPatra & Kurath, 2011). DNA vaccines designed based on other internal viral proteins of IHNV such as N, P, M, and NV did not induce any significant protective immunogenicity (Corbeil et al., 1999). Recently, an IHNV DNA vaccine encoding the G protein of the J genotype was found to be effective against a wide range of IHNV strains by eliciting strong neutralizing antibody responses and upregulation of Mx-1 gene, an IFN-inducible antiviral effector (Xu et al., 2017a). On other hand, DNA vaccines which consist of recombinant plasmid encoding the G protein derived from other serologically distant rhabdoviruses: SVCV or snakehead rhabdovirus (SHRV) were also shown to induce notable cross protections in the early (30 days post-vaccination) but not the late (70 days post-vaccination) lethal IHNV challenges (Kim et al., 2000). The G proteins of IHNV, SVCV, and SHRV shared only about 11% homology in amino acid sequences, therefore, protective responses observed during the early IHNV challenge could largely attributed to the non-specific innate immune responses conferred by IFN-inducible antiviral Mx-1 protein. As the non-specific immune responses faded over time, immunized fishes become more vulnerable to the late IHNV challenge, thus an increased mortality was observed. Nevertheless, fishes immunized with DNA vaccine encoding the G protein of IHNV survived in both the early and late IHNV challenges, suggesting that a long term effective protection requires specific immune responses (Kim et al., 2000). Previous studies have also indicated that co-infection and interactions between infectious pancreatic necrosis virus (IPNV) and IHNV have led to the loss of infective titer of IHNV due to the early release of interfering cytokines which inhibit the viral activities (Alonso, Rodriguez & Perez-Prieto, 1999; Saint-Jean & Perez-Prieto, 2007; Tafalla, Rodriguez Saint-Jean & Perez-Prieto, 2006). To investigate the capability of IPNV in inducing early cross protection against IHNV, De Las Heras, Perez Prieto & Rodriguez Saint-Jean (2009) created a DNA vaccine encoding the VP2 protein of IPNV, and demonstrated its protective efficacy against early heterologous IHNV challenges. Similar results were obtained when DNA vaccine against another rhabdovirus, VHSV was recruited for early IHNV challenge (LaPatra et al., 2001; Lorenzen et al., 2002b). However, the early non-specific cross protection conferred by the rhabdovirus DNA vaccines was shown to be restricted to viral but not bacterial infection as no increment in survival rate was detected when the immunized trout were challenged with bacterial pathogens (Lorenzen et al., 2002b).

Multivalent DNA vaccines

Viral hemorrhagic septicemia virus and IHNV are common pathogens endemic to rainbow trout in Europe. Co-administration of IHNV and VHSV DNA vaccines in a single injection in rainbow trout was previously reported to induce long-lasting protections against both individual and combined virus challenges (Boudinot et al., 1998; Einer-Jensen et al., 2009). Dual DNA vaccination could be a viable alternative to avoid repeated stressful vaccination procedures in rainbow trout. However, simultaneous vaccinations of several plasmid DNA encoding different antigens have also been reported to reduce the immunogenicity of the vaccines compared to those administered alone (Sedegah et al., 2004). Remarkably, a recent bivalent DNA vaccine encoding both the G protein of IHNV and VP2–VP3 of IPNV was shown to be highly effective in rainbow trout against individual and simultaneous IHNV and IPNV challenges. In all cases, the RPS was over 90% (Xu et al., 2017b). Multivalent vaccines have an added advantage over multi-DNA vaccination due to lower cost of production, as only one type of plasmid is required to produce multiple immunogenic antigens. However, the size of plasmid can affect its transformation into both the prokaryotic and eukaryotic cells (Kreiss et al., 1999; Ohse et al., 1995). Therefore, efforts should be given while designing DNA vaccines to minimize the size of recombinant plasmids, particularly those of multivalent vaccines. Studies on IHNV DNA vaccines are summarized in Table 2.

Table 2 Potential DNA vaccines produced and tested for IHNV.

Immunogens	Tested subject	Vaccination strategy	Outcome	Reference	
DNA encoding G protein of IHNV	Rainbow trout	IM, 100 ng	Protection against IHNV immersion (105 PFU/mL) challenges at 4 (RPS: 91.5%), and 7 (RPS: 93.5%) days post-vaccination, and IHNV IP (102 PFU in 50 µL) challenges at 28 (RPS: 91.5%), 120 (RPS: 86.5%) and 180 (RPS: 70%) days post-vaccination	Xu et al. (2017a)	
DNA encoding G protein of IHNV	Rainbow trout	IM, 1–100 ng	DNA vaccine dose of 1–10 ng conferred significant protections to the immunized fishes against IHNV IP challenge, and higher dose of DNA vaccine (100 ng) improved protection against a broad range of viral strains	LaPatra, Lorenzen & Kurath (2002)	
DNA encoding G protein of IHNV	Rainbow trout	IM, 10 μg	Protection against IHNV immersion (105 PFU/mL) challenges at 30 (RPS: 93%), and 70 (RPS: 87%) days post-vaccination	Kim et al. (2000)	
DNA encoding G protein of IHNV	Rainbow trout	IM, one μg	Protection against IHNV immersion (104 PFU/mL) challenge at 7 days post-vaccination (CM: 2%). When the immunized fishes were challenged with higher dose (105 PFU/mL) at 1–2 days post-vaccination, no significant protection was observed. However, the immunized fishes were partially protected (CM: ≈41%) when they were challenged at 4 days post-vaccination, and significantly protected when they were challenged at 7 days post-vaccination (CM: 20%)	LaPatra et al. (2001)	
DNA encoding G protein of IHNV	Rainbow trout	IM, one μg	Protection against IHNV immersion (105 PFU/mL) challenge at 18 days post-vaccination (CM: 18%)	Lorenzen et al. (2002b)	
DNA encoding G protein of IHNV	Rainbow trout	IM, IP, IB, GG, SS, 100 ng
Immersion treatment-water containing 3.4 × 106 DNA-coated magnetic polystyrene beads (10 mg of beads total weight). Concentration of DNA coated per mg beads weight was 60 μg	Fishes immunized via IM, IP, and GG route were protected (RPS: 100%, 50.3%, and 96.2%, respectively) from IHNV immersion (2.8 × 104 PFU/mL) challenges at 29 days post-vaccination. Vaccination via other routes did not induce significant protection against IHNV challenges	Corbeil, Kurath & LaPatra (2000)	
DNA encoding G protein of IHNV	Rainbow trout	IM, 0.001–5 μg	Strong protection (CM <6%) against homologous IHNV immersion challenges (101–104 PFU/mL) was observed in fishes immunized with DNA vaccine of various doses (0.1–5 μg). Significant protection (CM: 18%) was also induced in fishes immunized with DNA vaccine of as low as 0.001 μg.
Fishes immunized with 0.1 μg DNA vaccine were significantly protected from heterologous challenges including WRAC, RB-1, AK-14, 220-90, Shizuoka and 32–87 strains but not the Col-85 strain	Corbeil et al. (2000)	
DNA encoding G protein of IHNV	Rainbow trout	IM, 100 ng or 50 μg	Expression of G protein was detected in muscle, kidney, and thymus tissues, with levels peaking at 14 days and becoming undetectable by 28 days. No vaccine-specific pathological damage at the dose of 100 ng DNA per fish. Increased inflammatory response was observed when 50 μg DNA was administered	Garver et al. (2005)	
DNA encoding G protein of IHNV	Chinook salmon, sockeye salmon, kokanee salmon, rainbow trout	IM, 0.1 or 1 μg	DNA encoding G protein of IHNV protected Chinook and sockeye/kokanee salmon against IHNV immersion or IP challenge (RPS: 23–86 %) under variety of conditions but immunized rainbow trout was better protected (RPS: 100%)	Garver, LaPatra & Kurath (2005)	
DNA encoding G protein of IHNV	Rainbow trout	IM, 0.1–25 μg	DNA vaccine dose of one μg and above conferred complete protection to immunized fishes against IHNV IP (106 PFU per fish) challenge at 6 weeks post-vaccination	LaPatra et al. (2000)	
DNA encoding G protein of IHNV	Atlantic salmon, Rainbow trout	IM, 25 μg	Complete protection (RPS: 90–100%) against IHNV cohabitation (healthy fishes cohabitated with fishes injected with 4.9 × 103 PFU per fish) and immersion (4.6 × 103 PFU challenges at 8 weeks post-vaccination. Passive immunization of rainbow trout with immune serum from the immunized Atlantic salmon conferred significant protection against IHNV immersion challenge	Traxler et al. (1999)	
DNA encoding G protein of IHNV	Rainbow trout	IM, 0.1 μg	Complete protection against IHNV IP (103–108 PFU per fish) challenges in vaccinated fishes at 3 months post-vaccination. Viral challenges at 6, 13, 24, and 25 months post-vaccination showed protection with RPS values of 47–69%. No detectable histopathological damage due to DNA vaccination	Kurath et al. (2006)	
DNA encoding G protein of IHNV	Rainbow trout	IM, five μg	Expression of G protein was controlled by IRF1A promoter of fish origin, preventing its expression in human. Significant protection (CM: 19.4%) against IHNV immersion (105 PFU/mL) challenge in immunized fishes at 30 days post-vaccination	Alonso et al. (2003)	
DNA encoding G and M proteins of IHNV	Rainbow trout	IM, 1.5–5 μg	Fishes immunized with DNA encoding G (for protective immunity) and M proteins (apoptotic) of IHNV at various doses (1.5–5 μg) were significantly protected against IHNV immersion (105 PFU/mL) challenges. Vaccinated fishes that survived the challenge and received the ZnCl2 treatment at 30 days post-challenge demonstrated reduced G protein expression	Alonso, Chiou & Leong (2011)	
Poly(D,L-lactic-co-glycolic acid) (PLGA) nanoparticles containing DNA encoding G protein of IHNV	Rainbow trout	Oral route, 22 or 43 μg of DNA	Fishes immunized with low dose or high dose of nanoparticle containing the DNA were slightly protected against IHNV challenges at 6 and 10 weeks post-vaccination	Adomako et al. (2012)	
Alginate microsphere encapsulating DNA encoding G protein of IHNV	Rainbow trout	Immunization with (i) 10 μg DNA, (ii) 10 μg DNA then boosted once with same dose, (iii) 25 μg DNA, (iv) 25 μg DNA then boosted with same dose or (v) 100 μg via oral route	Alginate microsphere protected the encapsulated DNA vaccine from degradation in fish stomach and expression of G protein was detected in multiple tissue including gills, spleen, kidney, and intestinal tissues following vaccination. Expression of the genes related to innate and adaptive immune response increased with oral vaccine dose. Fishes immunized with 10, 20 (10 + 10), 25, 50 (25 + 25) or 100 μg DNA were partially protected from IHNV immersion (105 TCID50/mL) challenges at 30 days post-vaccination with RPS of 21%, 30%, 30%, 45%, and 56%, respectively	Ballesteros et al. (2015)	
DNA encoding G protein of U or M genotype of IHNV	Rainbow trout	IM, one μg	Fishes (1.2, 1.4, or 4 g) immunized with DNA encoding G protein of IHNV of M genotype were protected from homologous immersion (2 × 105 PFU/mL) challenges at 7 (RPS: 100%) and 28 (RPS: 88–100%) days post-vaccination. Similar protection level was observed against intraperitoneal (5 × 106 PFU/mL in 50 µL) IHNV (U genotype) challenge		
			Fishes immunized with DNA encoding G protein of IHNV of U genotype were protected from homologous intraperitoneal (5 × 106 PFU/mL in 50 µL) challenges at 7 (RPS: 86%) and 28 (RPS: 96%) days post-vaccination. Similar high protection level against immersion (2 × 105 PFU/mL) IHNV (M genotype) challenge was observed in bigger fishes (four g) but not juvenile fishes (1.2 g)	Peñaranda, LaPatra & Kurath (2011)	
DNA encoding N protein of IHNV	Rainbow trout	IM, one μg	Partial protection against IHNV immersion (104 PFU/mL) challenge at 28 days post-vaccination (CM: ≈38%). When the immunized fishes were challenged with higher dose (105 PFU/mL) at time points shorter than 1 week, no significant protection was observed	LaPatra et al. (2001)	
DNA encoding the N, P, M, NV or G protein of IHNV	Rainbow trout, sockeye salmon	IM, 1, 5, or 10 μg for rainbow trout. A total of 25 μg for sockeye salmon	Rainbow trout fry immunized with DNA encoding G protein at all doses were protected from immersion (105 PFU/mL) IHNV challenge (CM: 0–2%) at 4–6 weeks post-vaccination. Protection against IHNV reduced when these fishes were challenged with IHNV (IP, 106 PFU/mL in 100 µL) at 58 (CM: 31%) and 80 (CM: 49%) days-vaccination. DNA encoding other proteins induced no significant protections throughout the experiment
Passive immunization with immune sera from sockeye salmon immunized with DNA encoding G protein protected rainbow trout against IHNV immersion (105 PFU/mL) challenge (RPS: 100%)	Corbeil et al. (1999)	
Two immunogens, (i) DNA encoding G protein of IHNV and (ii) DNA encoding G protein VHSV	Rainbow trout	Co-administration of two immunogens via IM route, 30 μg each, boosted twice with same doses at 23 and 38 days after primary injection	Elicited IHNV and VHSV specific neutralizing antibodies following immunization. Activated Mx-gene and MHC class II expression at the site of injection. Immune responses induced in fishes by co-administration of the two immunogens were similar to those immunized separately	Boudinot et al. (1998)	
Two immunogens, (i) DNA encoding G protein of IHNV and (ii) DNA encoding G protein VHSV	Rainbow trout	Co-administration of two immunogens via IM route, one μg each	Protection against IHNV and VHSV immersion (1 × 104 TCID50/mL) challenges at 80 days post-vaccination (CM: 18%)	Einer-Jensen et al. (2009)	
DNA encoding G protein of SHRV	Rainbow trout	IM, 10 μg	Protection against early IHNV immersion (105 PFU/mL) challenge at 30 (RPS: 98%) days post-vaccination but not late immersion challenge at 70 (RPS: 26%) days post-vaccination	Kim et al. (2000)	
DNA encoding G protein of SVCV	Rainbow trout	IM, 10 μg	Protection against early IHNV immersion (105 PFU/mL) challenge at 30 (RPS: 95%) days post-vaccination but not late immersion challenge at 70 (RPS: 17%) days post-vaccination	Kim et al. (2000)	
DNA encoding VP2 of IPNV	–	–	BF-2 cells transfected with plasmid encoding VP2 induced an antiviral state against IPNV and IHNV infection	De Las Heras, Perez Prieto & Rodriguez Saint-Jean (2009)	
DNA encoding G protein of VHSV	Rainbow trout	IM, one μg	Protection against IHNV immersion (104 PFU/mL) challenges at 4, 7, and 14 (CM: 0–10%) days post-vaccination but not immersion challenge at 28 (CM: ≈69%) days post-vaccination	LaPatra et al. (2001)	
DNA encoding G protein of VHSV	Rainbow trout	IM, one μg	Protection against IHNV immersion (105 PFU/mL) challenge at 18 days post-vaccination (CM: 13%)	Lorenzen et al. (2002b)	
DNA encoding G protein of rabies virus	Rainbow trout	IM, one μg	No protection against IHNV immersion (104 PFU/mL) challenge	LaPatra et al. (2001)	
DNA encoding G protein of IHNV and VP2–VP3 gene of IPNV	Rainbow trout	IM, one μg	Protection against IHNV IP (102 PFU/mL in 100 µL) challenges at 30 (RPS: 93.3%), and 60 (RPS: 89.4%) days post-vaccination. Protection against simultaneous IHNV and IPNV IP (102 and 106 PFU/mL in 100 µL) challenges at 30 (RPS: 86.7%), and 60 (RPS: 92.3%) days post-vaccination	Xu et al. (2017b)	
Note:

P, phosphoprotein; M, matrix protein; NV, non-structural protein; N, nucleoprotein; G, glycoprotein; VP2, viral protein 2; VP3, viral protein 3; RPS, relative percentage of survival; CM, cumulative percentage mortality; IP, intraperitoneal delivery; IM, intramuscular delivery; IB, intrabuccal delivery; GG, gene gun delivery; SS, scarification of skin; TCID50, 50% tissue culture infectious dose; IRF1A, interferon regulatory factor 1A; PFU, plaque-forming unit; IHNV, infectious hematopoietic necrosis virus; SHRV, snakehead rhabdovirus; SVCV, spring viremia of carp virus; VHSV, viral hemorrhagic septicemia virus; IPNV: infectious pancreatic necrosis virus.

Factors affecting the efficacy of DNA vaccines

Protective immune responses induced by DNA vaccines could vary widely based on the route of immunization. Intramuscular injection is the most common DNA immunization technique employed in aquaculture, particularly fishes (Corbeil, Kurath & LaPatra, 2000; Garver, LaPatra & Kurath, 2005; Lorenzen et al., 2002a; Peñaranda, LaPatra & Kurath, 2011; Xu et al., 2017a). Corbeil, Kurath & LaPatra (2000) demonstrated that gene gun and intramuscular injection are the most efficient DNA delivery methods as measured by the protective efficacy on the immunized rainbow trout fry challenged with IHNV, whereas intraperitoneal injection induced partial protection. Nevertheless, other routes of DNA immunization including intrabuccal administration, scarification of the skin, and the immersion method were shown to be ineffective against IHNV challenge (Corbeil, Kurath & LaPatra, 2000). Although DNA vaccination via injection method is highly effective against IHNV, this technique is stressful to fishes, time consuming and laborious (Corbeil, Kurath & LaPatra, 2000).

A more cost-effective route of vaccination includes oral DNA vaccination. However, an oral vaccination requires the DNA to be protected from degradation in the digestive tract. Adomako et al. (2012) utilized a copolymer, poly(D,L-lactic-co-glycolic acid) as a nanocarrier for the delivery of oral DNA vaccine, where a slight protection towards immunized fishes was reported. Recently, Ballesteros et al. (2015) encapsulated the DNA encoding G protein of IHNV with an alginate microsphere, and orally vaccinated the rainbow trout. Their results revealed that the DNA vaccine was effectively protected in the fish gut by the alginate microsphere, resulting in a significant reduction in mortality of the immunized fishes. To date, oral vaccination is less effective compared to intramuscular injection. However, further optimization in the future could possibly enhance the protective efficacy of these vaccines. Therefore, it represents a viable alternative in aquaculture, in which it is more practical: lower cost and less laborious.

DNA vaccine delivery by attenuated bacteria via horizontal gene transfer was also previously suggested to be a suitable route of immunization in aquaculture due to its low labor cost. Despite successful demonstration of GFP gene transfer into salmonid fish cells via attenuated invasive E. coli, in vitro or in vivo gene transfer of IHNV G protein into fish cells has not been conducted (Simon & Leong, 2002).

The efficacy of IHNV DNA vaccines has been reported to be dose dependent (Ballesteros et al., 2015; Corbeil et al., 2000; Garver, LaPatra & Kurath, 2005; LaPatra et al., 2000). In general, a larger fish requires a higher vaccination dose for effective protection against IHNV. A 120 g-fish requires about 100 times higher dosage to achieve similar protective immunity compared to fingerlings of one to three g (LaPatra et al., 2000). LaPatra, Lorenzen & Kurath (2002) later demonstrated that 0.1 µg of IHNV DNA vaccine is sufficient to induce significant protection in rainbow trout fry. As a rule of thumb, to induce sufficient protective immune response in rainbow trout, intramuscular vaccination of at least 10 ng DNA per gram body weight is required (Lorenzen et al., 2002a).

IHNV DNA vaccines are most commonly tested on rainbow trout, often resulting in high neutralizing antibodies and survival rate in the fish (Corbeil et al., 1999, 2000; Kim et al., 2000; LaPatra, Lorenzen & Kurath, 2002; Xu et al., 2017a). Atlantic salmon was also recruited as an animal model to study the efficacy of IHNV DNA vaccine, in which they were greatly protected from IHNV immersion and cohabitation challenges (over 90% RPS). Furthermore, passive serum transfer from the immunized Atlantic salmon to rainbow trout has also increased the survival rate of the recipients (Traxler et al., 1999). On the other hand, Chinook and sockeye salmon immunized with DNA vaccines also exhibited increased survivability, although to a lesser extend compared to Atlantic salmon and rainbow trout (Garver, LaPatra & Kurath, 2005).

Apart from host differences, external parameters such as temperature also play an important role in determining the efficacy of the DNA vaccines. Lorenzen et al. (2002a) suggested that the DNA vaccine encoding G protein of rabies virus failed to elicit early unspecific protection against IHNV could be due to the low water temperature. Intriguingly, Lorenzen et al. (2009) later demonstrated that IHNV and VHSV DNA vaccines induced different defense mechanisms in rainbow trout upon VSHV challenge at different temperatures. At low temperature of 5 and 10 °C, IHNV DNA vaccine could induce a prolonged cross protection against VSHV challenge but no significant protection was observed at 15 °C. In addition, the activity of Mx protein and the level of neutralizing antibody of the immunized fish were also found to vary at different temperatures (Lorenzen et al., 2009). Therefore, the effect of vaccines at different water temperatures should be studied to achieve an optimal protection.

Vaccine efficacy can be affected by the route of vaccine delivery as different vaccination approaches influence vaccine localization and priming of the immune cells, and consequently affect the systemic immune responses (Zhang, Wang & Wang, 2015). Due to the complexity of different vaccines, hosts, vaccine dosages, types of adjuvant involved, injection volumes and intervals between injections, thus relative immunogenicity of the vaccines administered by different routes could vary considerably (Zhang, Wang & Wang, 2015). The underlying mechanism of different routes of vaccination in affecting DNA vaccine’s efficacy in fishes remains elusive. Nevertheless, intramuscular injection is the most widely used method for DNA vaccination in fishes due to its ability to induce potent immune responses (Tonheim, Bogwald & Dalmo, 2008). Studies in mice demonstrated the distribution of plasmid DNA between the muscle body and epimysium following a DNA vaccination, subsequently myocytes and mononuclear cells were shown to rapidly uptake plasmid DNA shortly after intramuscular injection (Hølvold, Myhr & Dalmo, 2014). DNA immunization by gene gun, on the other hand, introduced the DNA plasmid directly into the cytoplasm, presumably resulting in the DNA being processed by antigen presenting cells, and subsequently activating the adaptive immunity (Wang et al., 2008). DNA vaccines delivered via oral route are relatively less laborious but they were shown to be less effective. Oral DNA vaccination required special protection for the plasmid DNA against hostile fish digestive system to prevent DNA degradation before cellular uptake. Even the DNA plasmid was protected from degradation via certain approaches, transfection efficiency of the plasmid DNA in the fish digestive system poses another challenge (Corbeil, Kurath & LaPatra, 2000). Immersion route is simple and suitable for mass vaccination of fishes. However, transfection efficiency of the plasmid DNA delivered via immersion route is heavily affected by many factors such as the length of immersion time, size of the fish, stress, pH, osmolarity of the vaccine buffer, the water temperature, and the physical properties (particulate or soluble) of the antigen. Each parameter has to be optimized to improve transfection efficiency and immunogenicity of the DNA vaccine (Nakanishi & Ototake, 1997).

Controversial in DNA vaccination

Despite the tremendous amount of promising results yielded by DNA vaccines against fish pathogens, the introduction of foreign DNA into human foods has always been controversial throughout the past decades (Alonso & Leong, 2013). There is a possibility that the plasmid DNA could integrate into the host genome, leading to insertion mutations. Nevertheless, plasmid DNA delivered via intramuscular injection into muscle cells exists as an extra-chromosomal DNA, and its integration into the host genome was reported to be negligible (Kanellos et al., 1999; Ledwith et al., 2000; Nichols et al., 1995). To further mitigate this issue, Alonso et al. (2003) developed a DNA vaccine based on the G gene of IHNV, controlled by the interferon regulatory factor 1A promoter originated from rainbow trout to prevent its expression in human. In addition, a study by Salonius et al. (2007) also indicated that the potential risk of spontaneous mutations in Atlantic salmon was about 43-folds higher than that caused by DNA vaccination. Furthermore, several studies have suggested that IHNV DNA vaccination of rainbow trout only caused transient histopathological changes in multiple tissues, and no long-term histopathological damage was observed (Garver et al., 2005; Kurath et al., 2006). However, certain regulations such as the Norwegian Gene Technology Act which categorized DNA vaccinated animals as genetically modified organisms presents a stringent policy, eventually leading to low public acceptance (Alonso, Chiou & Leong, 2011). To eliminate this concern, a self-destructive IHNV DNA vaccine was designed (Alonso, Chiou & Leong, 2011). The plasmid DNA contains an inducible fish cell promoter which regulates the expression of G glycoprotein for protective immune responses, and a ZnCl2 inducible promoter which controls the expression of IHNV M protein inducing apoptosis of the transfected cells. Upon successful vaccination, fishes were significantly protected from lethal IHNV challenge, and exposure to ZnCl2 induced apoptosis in fish cells containing the DNA vaccine without causing serious toxicity to the fishes (Alonso, Chiou & Leong, 2011). This approach could pave a way to the development of safer DNA vaccines with higher public acceptance.

Although many research groups have patented their inventions, including Kurath et al. (1985) (Patent No.: 5354555), Salonius et al. (2007) (Patent No.: EP1553979A1, CA2498896C, CN100339131C, JP4578973B2, ES2288627T3, DK1553979T3, DE60315858T2, PT1553979E, AT370746T, AU2003277863B2, WO2004026338A1, NO20051840L, HK1082666A1, CY1107784T1), Alonso et al. (2003) and Alonso, Chiou & Leong (2011) (Patent No.: WO/2002/069840), and Xu et al. (2017a) (Patent No.: CN105816871A, CN105861450A), to date, the Apex-IHN® manufactured by Aqua Health Ltd (an affiliate of Novartis) is the only licensed IHNV DNA vaccine in Canada and the USA (Grunwald & Ulbert, 2015; USDA, 2014).

Conclusions

Up until now, no effective treatment is available for fishes infected by IHNV. Apart from good biosecurity measures, immediate isolation of symptomatic fishes, rapid and accurate diagnosis of IHNV followed by culling of the infected fishes are essential to prevent the virus from spreading to other farm sites, and possibly prevent a farm-wide infection. A combination of both the rapid on-site test (staphylococcal coagglutination test or RT-LAMP) for mass sample screening, and laboratory confirmatory tests (ELISA and RT-PCR) should be performed to achieve a balance between speed and accuracy. Vaccination provides an alternative approach for fish farmers who can afford extra costs to protect their fishes from IHNV infection. As potentially low-cost vaccines such as oral vaccines have yet to show promising results, vaccination may not be applicable to farmers with small capital in the near future. To date, Apex-IHN® is the only licensed DNA vaccine approved in Canada and the USA. Despite its outstanding protective efficacy, the use of DNA vaccine is still very limited at the moment. Hence, studies focusing on the safety of DNA vaccines should be encouraged.

Additional Information and Declarations

Competing Interests

Author Contributions

Data Availability

The authors declare that they have no competing interests.

Chean Yeah Yong conceived and designed the contents of the paper, performed data searches, analyzed the data, prepared figures and/or tables, authored or reviewed drafts of the paper, approved the final draft.

Hui Kian Ong conceived and designed the contents of the paper, performed data searches, analyzed the data, prepared figures and/or tables, authored or reviewed drafts of the paper, approved the final draft.

Hooi Chia Tang conceived and designed the contents of the paper, performed data searches, analyzed the data, prepared figures and/or tables, authored or reviewed drafts of the paper, approved the final draft.

Swee Keong Yeap conceived and designed the contents of the paper, performed data searches, analyzed the data, prepared figures and/or tables, authored or reviewed drafts of the paper, approved the final draft.

Abdul Rahman Omar conceived and designed the contents of the paper, analyzed the data, authored or reviewed drafts of the paper, approved the final draft.

Kok Lian Ho conceived and designed the contents of the paper, analyzed the data, authored or reviewed drafts of the paper, approved the final draft.

Wen Siang Tan conceived and designed the contents of the paper, analyzed the data, authored or reviewed drafts of the paper, approved the final draft.

The following information was supplied regarding data availability:

No raw data has been generated from the literature review.

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
