# Peer review of "Infectious hematopoietic necrosis virus: advances in diagnosis and vaccine development"

_PeerJ, doi:10.7717/peerj.7151_

## Round 0.1 · original submission · Major Revisions

Though both the reviewers comment that the manuscript is well written, one of the reviewers raise a lot of queries on various aspects of IHNV and vaccine development, especially validity of the findings; and need major revisions. In my opinion, the manuscript is a very good piece but please give the necessary explanations on the reviewers comments. Thank you!

Reviewer 1 ·

Basic reporting

Overall, The article by Yong et al is well written . However, I detected some english misspellings such as the word "salmons". I recommend the authors to check potential misspellings . Sufficient backgroud is provided and the introduction adequately introduces the subject The review is of cross-disciplinary interest since recombinant IHNV vaccines can be used to control infectious diseases in many different animal species. Diagnosis of salmonid fish was previosuly reviewed in the book "Molecular diagnosis of salmonid fish" edited by Carey Cunningham (Springer-Science) and in the book "Fish Diseases and Disorders" edited by Woo et al., 2011. However, to my knowledge the area of the development of vaccines against IHNV has not been deeply reviewed. This review offers a complete picture of all the vaccines developed against IHNV. I particularly find figure 1 very interesting.

I have missed some vaccines in this review. For instance Alonso et al, 2003 developed the first fish specific expression vector containing the interferon regulatory factor 1A (IRF1A) promoter for genetic immunization of fish. The same autors developed a suicidal DNA vaccine for infectious hematopoietic necrosis virus (IHNV) in 2011. Both references should be included in Table 1 and the first one disccused in the mansucript.

In addition the authors shoud specify which vaccines from Table 1 have been patented, which are commertially available and which countries are currently using vacines to control IHNV infections in fish farms and under which conditions. Killed vaccines and a DNA vaccine have been licensed for commercial use in Atlantic salmon net-pen aquaculture on the west coast of North America where such vaccines can be delivered economically by injection. However, vaccines against IHNV have not yet been licensed in other countries where the application of vaccines to millions of smaller fish will require additional research on novel mass delivery methods. Are IHNV vaccines being used in Chile and Norway?

I believe that the survey methology section should be deleted .

Line 218 should be clarified. Please give more deaeils about the mutant tested by Roberti et al., 1998.

Concluision section should be revised

Experimental design

The article content is within the aims and scope of the journal.
The authors provide a full coverage of the subject. Sources were adequately cited.
I encourage the authors to organize the review in three subsections: General introduction, Diagnosis of IHNV and Vaccines against IHNV. Line 97 should be include in the clinical diagnosis section.

Validity of the findings

- In the conlusions sections I believe that the autors should emphize the OIE recommendations for IHNV diagnosis and its definition of a suspected or confimed case. Please check http://www.oie.int/fileadmin/Home/eng/Health_standards/aahm/current/chapitre_ihn.pdf. The methods included in the conclusions do not agree with the OIE recommendations.

- Line 307 -the authors state that multivalent DNA vaccines have an added advantage over multi-DNA vaccination due to its lower cost of producion, as only one type of plasmid is required to produce multiple immunogenic antigens. However, it has been demonstrated that the size of the plasmid might affect the efficacy of the plasmid transfection into muscle cells. The aurthors should disccuss this issue, since the size of the plasmid can affect the efficacy of the DNA vaccines.

Line 315- can the authors explain why the admistration route affects the efficay of the DNA vaccines.?

Line 32- Line 394, conclusions. The authors state that rapid and accurate diagosis are important to preven IHNV outbreaks. If IHNV is detected in a farm, what is the next step. Is this disease noticeable? If IHNV is detected, are all the fish in the fish farm killed? are they separated? Do the authors really belive that IHNV diagnosis helps to prevent the outbreak?. or is diagnosis a control measure instead?

Line 401- Is apex-IHNV being used in smal farms in Canada?

-The thorough disinfection of fertilised eggs, the use of virus-free water supplies for incubation and rearing, and the operation of facilities under established biosecurity
measures are all critical for preventing IHN at a fish production site. Biosecurity measures are important to prevent IHNV outbreaks and should be detailed in the text.

·

Basic reporting

no comment

Experimental design

No comment

Validity of the findings

No comment

Additional comments

The manuscript is well written, and fully covers main progresses on diagnosis technologies and vaccines development issues of IHNV. However, there are minor concerns should be verified before publishing.

1. The sentence in Line 44-45 need to cite a reference.
2. I think the sentence in line 45-47 is describing the general mortality and economic caused by IHNV worldwide. If it is true, please cite a review of IHNV instead of the paper just about IHN outbreak in Iran. Or add more references.
3. In the section “Diagnosis of IHNV”, references should be cited to support the abnormal behaviors in infection of IHNV. The subtitle “Diagnosis of IHNV” is not appropriate due to the fact that this part is just diagnosis of IHNV by behaviors.
4. Sentence in line 223-224 is ambiguous and lacks supportive references. Although there are several studies about attenuated vaccine against IHNV, but nothing to do with commercialization.
5. Reference in line 295 is not correct.
6. Sentence in line 387-388 “Apart from live attenuated vaccines, Apex-IHN ® is the only licensed DNA vaccine approved in Canada.” The sentence is ambiguous. It seems that there are commercialized live attenuated vaccines against IHNV. But the Apex-IHN is the only one commercialized vaccine against IHNV.

---

## Round 0.2 · accepted · Accept

Congratulations!
Thank you for the clarification of raised queries!
Good luck for your next attempt!
Cheers!

# Reviewer 1 ·

Basic reporting

No comment

Experimental design

No comment

Validity of the findings

No comment

Additional comments

The authors have correctly addressed my previous concerns